# New Benzimidazothiazolone Derivatives as Tyrosinase Inhibitors with Potential Anti-Melanogenesis and Reactive Oxygen Species Scavenging Activities

**DOI:** 10.3390/antiox10071078

**Published:** 2021-07-05

**Authors:** Hee Jin Jung, Dong Chan Choi, Sang Gyun Noh, Heejeong Choi, Inkyu Choi, Il Young Ryu, Hae Young Chung, Hyung Ryong Moon

**Affiliations:** College of Pharmacy, Pusan National University, Busan 46241, Korea; hjjung2046@pusan.ac.kr (H.J.J.); tjdahdir@pusan.ac.kr (D.C.C.); rskrsk92@pusan.ac.kr (S.G.N.); heejeong@pusan.ac.kr (H.C.); godhot785@pusan.ac.kr (I.C.); iy102@pusan.ac.kr (I.Y.R.)

**Keywords:** tyrosinase inhibitor, reactive oxygen species (ROS), melanogenesis, benzimidazothiazolone

## Abstract

Thirteen (*Z*)-2-(substituted benzylidene)benzimidazothiazolone analogs were synthesized and evaluated for their inhibitory activity against mushroom tyrosinase. Among the compounds synthesized, compounds **1**–**3** showed greater inhibitory activity than kojic acid (IC_50_ = 18.27 ± 0.89 μM); IC_50_ = 3.70 ± 0.51 μM for **1**; IC_50_ = 3.05 ± 0.95 μM for **2**; and IC_50_ = 5.00 ± 0.38 μM for **3**, and found to be competitive tyrosinase inhibitors. In silico molecular docking simulations demonstrated that compounds **1**–**3** could bind to the catalytic sites of tyrosinase. Compounds **1**–**3** inhibited melanin production and cellular tyrosinase activity in a concentration-dependent manner. Notably, compound **2** dose-dependently scavenged ROS in B16F10 cells. Furthermore, compound **2** downregulated the protein kinase A (PKA)/cAMP response element-binding protein (CREB) and mitogen-activated protein kinase (MAPK) signaling pathways, which led to a reduction in microphthalmia-associated transcription factor (MITF) expression, and decreased tyrosinase, tyrosinase related protein 1 (TRP1), and TRP2 expression, resulting in anti-melanogenesis activity. Hence, compound **2** may serve as an anti-melanogenic agent against hyperpigmentation diseases.

## 1. Introduction

Tyrosinase (polyphenol oxidase, EC 1.14.18.1), a binuclear copper-containing monooxygenase, is a critical rate-limiting melanogenic enzyme involved in melanogenesis, the process of melanin synthesis in the skin. Melanogenesis is initiated by the hydroxylation of tyrosine to l-3,4-dihydroxy-phenylalanine (DOPA), which is catalyzed by tyrosinase. Tyrosinase also catalyzes the subsequent enzymatic conversion of DOPA to dopaquinone [1,2]. Tyrosinase is the key factor involved in inducing dermatological disorders, including age spots, freckles, and melasma. Commercial tyrosinase inhibitors, such as hydroquinone, arbutin [3], kojic acid [4], ellagic acid [5], and tranexamic acid, have been used as skin-whitening agents; however, they are associated with certain side effects, including carcinogenicity, chemical instability, and poor bioavailability [6,7]. Thus, novel and efficient, anti-tyrosinase inhibitor with a favorable safety profile are necessary for anti-hyperpigmentation. There is a need to develop safe skin-whitening agents in order to overcome the limitations of established products.

α-Melanocyte-stimulating hormone (α-MSH) and 3-isobutyl-1-methylxanthine (IBMX) are the main physiological inducers of melanogenesis and accelerate tyrosinase activity through the cyclic adenosine monophosphate (cAMP) signaling pathway [8]. Furthermore, α-MSH binds to melanocortin-1 receptors (MC1R) on the surface of melanocytes to activate the protein kinase A (PKA) pathway, phosphorylate the cAMP response element-binding protein (CREB) transcription factor; and induce the expression of microphthalmia-associated transcription factor (MITF), thus promoting melanogenesis [9,10]. Activated MITF induces the expression of melanogenic enzyme genes, such as tyrosinase-related protein 1 (TRP1) and dopachrome tautomerase (TRP2; DCT) [11,12,13]. The PKA/CREB and mitogen-activated protein kinase (MAPK) signaling pathways are important in melanogenesis; therefore, we wondered if chemicals or agents that block or activate this axis could have anti- or pro-melanogenic activity and thus serve as potential therapeutic agents for hyperpigmentation disorders [14,15,16].

Reactive oxygen species (ROS) have been linked to several disorders, including aging and age-related diseases [17,18]. Importantly, ROS levels can accelerate skin aging and increase both melanocyte proliferation and hyperpigmentation [19,20]. Among the reactive species (RS) derived from melanocytes and keratinocytes, nitric oxide (∙NO) and superoxide anion radical (O_2_˙^−^) stimulate melanin synthesis by enhancing the protein expression of tyrosinase and TRP1 [21,22,23,24]. Therefore, antioxidants, inhibitors of tyrosinase, and scavengers of ROS may suppress melanogenesis in the epidermis.

Benzimidazothiazoles, as core rings, are polycyclic isosteres whose derivatives have been reported to exert anticancer [25] and anti-inflammatory [26,27] activities. However, no study has yet described its anti-tyrosinase and hyperpigmenting effects in vitro model. Previously, we reported that (*Z*)-β-phenyl-α,β-unsaturated carbonyl scaffolds inhibit tyrosinase and melanogenesis both in vitro [28,29,30,31]. As our continuous efforts to find potent tyrosinase inhibitors using (*Z*)-β-phenyl-α,β-unsaturated carbonyl scaffolds, we integrated the benzimidazothiazolone template with other benzaldehydes to synthesize a (*Z*)-β-phenyl-α,β-unsaturated carbonyl scaffold.

In the present study, we evaluated the ability of (*Z*)-2-(substituted benzylidene)benzimidazothiazolone derivatives to inhibit mushroom tyrosinase enzyme, and performed kinetic studies and molecular docking simulations in silico; and investigated the scavenge ROS and the hyperpigmentation mechanism of a potent compound expression levels of melanogenic proteins and mRNA in B16F10 cells. As part of our continued work on potent tyrosinase inhibitors, this study aimed to evaluate the anti-melanogenic effects of benzimidazothiazolone derivatives.

## 2. Materials and Methods

### 2.1. Reagents and Antibodies

Dulbecco’s modified Eagle’s medium (DMEM) and fetal bovine serum (FBS) were purchased from WelGene (Gyeongsan-si, Daegu, Korea). Kojic acid, l-tyrosine, l-DOPA, mushroom tyrosinase (EC 1.14.18.1), α-melanocyte-stimulating hormone (α-MSH), and 3-isobutyl-1-methylxanthine (IBMX) were purchased from Sigma-Aldrich (St. Louis, MO, USA). Antibodies against tyrosinase (sc-7833), MITF (sc-11002), phospho-CREB (p-CREB) (sc-81486), p-PKA (sc-12905), PKA (sc-365615), p-JNK (sc-6254), p-ERK (sc-7383), ERK (sc-514302), p-p38 (sc-7973), p38 (sc-535), TFIIB (sc-271736), and β-actin (sc-47778) were obtained from Santa Cruz Biotechnology (Santa Cruz, CA, USA). CREB (#9197s) and JNK (#9252s) antibody purchased from Cell Signaling Technology (Danvers, MA, USA). Peroxidase-labeled anti-rabbit, anti-mouse, and anti-goat immunoglobulins were acquired from GeneTex (Irvine, CA, USA). Reagent-grade chemicals and solvents were purchased from commercial sources. Ultra-pure water was used throughout the experiment.

### 2.2. General Experimental Methods

#### 2.2.1. Mushroom Tyrosinase Inhibitory Assay

The tyrosinase inhibitory activity assay was accomplished using mushroom tyrosinase enzyme, as previously described by Hyun et al., 2008 [32], with minor modifications. The IC_50_ (concentration necessary for 50% inhibition of enzyme activity) was calculated by constructing a linear regression curve presenting compound concentrations on the *x*-axis and percentage inhibition on the *y*-axis. A negative control was obtained by adding dimethyl sulfoxide (DMSO) instead of the compounds. Kojic acid was used as a positive control.

#### 2.2.2. Enzyme Kinetic Analysis of Mushroom Tyrosinase

Lineweaver–Burk and Dixon plots [33,34] were used to calculate the inhibition mode and inhibition constant (*K*_i_) against mushroom tyrosinase. Kinetic parameters for various concentrations of the l-tyrosine substrate (0.5–4 mM) and inhibitors (0.4, 2, and 10 µM for compound **1**; and 2, 4, and 8 µM for compound **2**; and 2.5, 5, and 10 µM for compound **3**, respectively) were achieved.

#### 2.2.3. Mushroom Tyrosinase Molecular Docking Simulations

The AutoDock4.2 software (La Jolla, CA, USA) was used to determine the structure of the enzyme-inhibitor complex and confirm precision, repeatability, and reliability of the docking results. For docking studies, the crystal structure of mushroom tyrosinase protein target was obtained from protein sequence alignment (PDB ID: 2Y9X) (http://www.rcsb.org/adb, accessed on 10 February 2021) [35]. For the docking procedure, the 2D structure was converted into a 3D structure, charges were planned, and hydrogen atoms were added using the ChemDraw program (ChemDraw Ultra 10.0, Cambridge soft) (http://www.cambridgesoft.com, accessed on 12 February 2021) [36]. The docking results were visualized and analyzed using the UCSF Chimera molecular graphics system (v1.14, University of California, Berkeley, CA, USA) and AutoDockTools (v1.5.6, The Scripps Research Institute, La Jolla, CA, USA).

### 2.3. Biological Evaluation

#### 2.3.1. Cell Culture and Viability Analysis

The mouse melanoma cell line B16F10 was acquired from American Type Culture Collection (ATCC^®^; CCL-6475™) (Manassas, VA, USA) and cultured in DMEM, supplemented with 10% fetal bovine serum (FBS) and 1% penicillin–streptomycin, at 37 °C in a humidified 5% CO_2_ atmosphere. Cell viability assays were performed with minor modifications, as previously reported [37]. Cell viability was evaluated using the EZ-Cytox assay kit (DaeilLab, Seoul, Korea). The compounds were dissolved in DMSO for the in vitro experiments.

#### 2.3.2. Determination of Intracellular ROS Levels

ROS scavenging activity was determined according to the method reported by Ali et al. [38] and Lebel and Bondy [39]. Briefly, 2′,7′-Dichlorodihydrofluorescein-diacetate (DCFH-DA) (2.5 mM, Molecular Probes, Eugene, OR, USA) mixed with esterase (1.5 units/mL), was incubated at 37 °C for 30 min and placed on ice in the dark until immediately prior to the study. Phosphate buffer (50 mM) at pH 7.4 was used. To determine intracellular ROS scavenging activity, B16F10 cells (1 × 10^4^/well) were seeded in black 96-well plates. After 24 h, the cells were incubated with DCFH-DA (20 µM) and SIN-1 (200 µM) for 30 min to induce ROS generation. The fluorescence intensity of oxidized DCFH was measured using a fluorescent microplate reader at excitation and emission wavelengths of 485 and 530 nm, respectively (Tecan, Männedorf, Switzerland). Trolox was used as the positive control.

#### 2.3.3. Melanin Contents in B16F10 Cells

The effect of inhibitors on the melanin content in B16F10 cells was investigated as previously described [40,41]. The amount of melanin was determined spectrophotometrically by measuring the absorbance at 492 nm using a microplate reader (Tecan, Männedorf, Switzerland). All results were standardized to the total protein concentration of the cell lysates using the bicinchoninic acid (BCA) reagent (Thermo Fisher Scientific, Waltham, MA, USA).

#### 2.3.4. Cellular Tyrosinase Activity Assay in B16F10 Cells

B16F10 cells were seeded at a density of 3 × 10^3^/well in 6-well plates for 24 h at 37 °C in an atmosphere of 95% air/5% CO_2_. Each test agent was dissolved in DMSO at a final concentration up to 20 µM and pretreated with the indicated concentration of each compound; then, α-MSH (5 µM) and IBMX (200 µM) were added to the medium for 48 h. After treatment, the cells were washed with ice-PBS and lysed by the addition of 50 mM phosphate buffer containing 1% Triton X-100 and 1% PMSF at −80 °C for 1 h. The lysates were clarified by centrifugation at 5600× *g* for 10 min at 4 °C; then, 80 µL of the cell lysate was added to 20 µL of l-DOPA (2 mg/mL in distilled water). The mixture was incubated for 10 min at 37 °C, and the absorbance of the reaction mixture at 492 nm was recorded using a microplate reader (Tecan, Männedorf, Switzerland). Kojic acid was used as the positive control.

#### 2.3.5. RNA Isolation and Quantitative Real-Time PCR (qPCR)

RNA from cell samples was purified using the RiboEx Total RNA solution (GeneAll Biotechnology, Seoul, Korea) according to the manufacturer’s instructions. Total RNA (2 µg) treated with ribonuclease (RNase) free deoxyribonuclease (DNase) was reverse-transcribed using Hyperscript^TM^ One-Step RT-PCR (GeneAll Biotechnology, Seoul, Korea). Real-time qPCR was performed using the SensiFAST^TM^ SYBR^®^ No-ROX dye (Bioline, London, UK.) and a CFX Connect System (Bio-Rad Laboratories, Hercules, CA, USA). Relative gene expression was calculated with the standard curve method using *GAPDH* as the internal control. Ct values were obtained with qPCR. Three independent tests were performed. The primer sequences are displayed in Table 1.

#### 2.3.6. Western Blot Analysis

Lysed samples were boiled for 10 min with gel-loading buffer (0.125 M Tris-HCl, pH 6.8, 4% SDS, 10% 2-mercaptoethanol and 0.2% bromophenol blue) at a volume ratio of 1:1. Total protein equivalents were separated with SDS-PAGE using 9–10% acrylamide gels and then transferred onto polyvinylidene fluoride membranes (Millipore, Burlington, MA, USA) at 25 V for 10 min in a semi-dry transfer system (Bio-Rad Laboratories, Hercules, CA, USA). The membranes were immediately placed in blocking buffer (5% non-fat milk) in 10 mM Tris (pH 7.5), 100 mM NaCl, and 0.1% Tween-20. The blots were blocked at room temperature for 1 h. The membranes were incubated with appropriate specific primary antibodies at 4 °C overnight and then treated with horseradish peroxidase-conjugated anti-mouse, anti-rabbit or anti-goat antibodies (1:5000) at 25 °C for 1 h (Santa Cruz Biotechnology, Dallas, TX, USA). Protein bands were visualized using the SuperSignal^®^ West Pico Chemiluminescent Substrate kit (Advansta, San Jose, CA, USA) and Davinch-Chemi^TM^ (Davinch-K, Seoul, Korea).

### 2.4. Statistical Analysis

Data are expressed as mean ± standard error of the mean (SEM). The statistical significance of the differences between groups was determined with one-way ANOVA followed by Dunnett’s test. An associated probability of (*p* value) < 0.05 was considered significant.

## 3. Results and Discussion

### 3.1. Synthesis of (Z)-2-(Substituted benzylidene)benzimidazothiazolone Derivatives ***1**–**13***

The synthetic route for the benzimidazothiazolone derivatives **1**–**13** is illustrated in Scheme 1. To synthesize the core template benzimidazothiazolone **15**, commercially available 2-mercaptobenzimidazole was first condensed with bromoacetic acid. The reaction provided the desired compound **15** at a yield of 8%, with an intermediate **14** (59%). In the presence of acetic anhydride and pyridine, the intermediate **14** could be converted to the core template **15** rapidly and in high yield (91%) under reflux. Heating **15** and various benzaldehydes in the presence of acetic acid and sodium acetate for 15 h to 3 days produced 13 benzimidazothiazolone derivatives **1**–**13** as solids in yields of 35–82%. The configuration of the newly formed double bond was determined using the vicinal ^1^H and ^13^C-coupling constants (^3^*J*) in the proton-coupled ^13^C spectra.

Vögeli et al. [42] reported that the configuration of trisubstituted exocyclic C, C-double bonds in a general structure A (Figure 1) differentiated the C, H-spin coupling constants over three bonds. The ^3^*J* of C(1) of compounds in which the oxygen of carbonyl and the 3-hydrogen were on the same side was 3.6 to 6.4 Hz, whereas that of the compounds in which the oxygen of carbonyl and 3-hydrogen were on the same side was roughly twice as large (generally >10 Hz). The ^3^*J* of C(1) of compound **11** was 5.3 Hz, confirming that **11** is a (*Z*)-isomer. Structures of the 13 final compounds were confirmed with ^1^H and ^13^C NMR spectroscopy and mass spectroscopy.

### 3.2. Inhibitory Activities of (Z)-2-(Substituted Benzylidene)benzimidazothiazolone Derivatives ***1**–**13*** against Mushroom Tyrosinase

To choose potent derivatives for cell-based in vitro of anti-tyrosinase and melanogenic-inhibitory activity, we synthesized 13 (*Z*)-2-(substituted benzylidene)benzimidazothiazolone derivatives **1**–**13**, which were determined using mushroom tyrosinase and kojic acid (as a potent tyrosinase inhibitor) [43,44]. The inhibitory potential of compounds **1**–**13** against tyrosinase was assessed using l-tyrosine as the substrate, and the results were described as the IC_50_ values determined with linear regression analysis (Table 2). Of the 13 synthesized compounds, three exhibited potent tyrosinase inhibitor; **1** (IC_50_ = 3.70 µM) with a 4-hydroxy substituent; **2** (IC_50_ = 3.05 µM) with a 3,4-dihydroxy substituent; and **3** (IC_50_ = 5.00 µM) with a 2,4-dihydroxy substituent. Specifically, the inhibitory potencies of compounds **1**, **2**, and **3** were 4.9-, 6.0-, and 3.7-fold greater, respectively, than those of kojic acid (IC_50_ = 18.27 µM). Compounds **4**, **5**, **8**, **10**, **11**, and **13** showed moderate inhibitory activities, with IC_50_ values of 42.06 µM, 21.03 µM, 41.40 µM, 27.96 µM, 34.10 µM, and 34.60 µM, respectively. Compounds **6**, **7**, **9**, and **12** were inactive at the tested concentrations.

Insertion of an additional hydroxyl group into the β-phenyl ring of compound **1** with a 4-hydroxyphenyl ring increased tyrosinase inhibitory activity (compounds **1** vs. **2** and **3**). Furthermore, the introduction of methoxyl, ethoxyl, or bromo groups at the 3-position of the β-phenyl ring of compound **1** with a 4-hydroxyphenyl ring abated tyrosinase inhibitory activity (compounds **1** vs. **4** and **5** and **13**). Compounds **7**, **8**, **9**, and **11,** without a hydroxyl group on the β-phenyl ring showed weak or no inhibition. The 3-methoxy-4-hydroxy group of the β-phenyl ring increased tyrosinase inhibitory activity compared to the 3-hydroxy-4-methoxy group of the β-phenyl ring (compounds **4** vs. **6**). Unlike the hydroxyl group, 2,4-dimethoxyphenyl groups act as hydrogen bond acceptors and are very weak or inactive. The present results are in close agreement with those of a previous reported [45].

Based on the results of our previous studies, 3,4-dihydroxyphenyl compound exhibit potent inhibitory activity against mushroom tyrosinase and cell-based cellular tyrosinase assays [46]. As expected, compound **2**, with a 3,4-dihydroxyphenyl substituent (catechol moiety), exerted excellent tyrosinase inhibition, and compounds with 4-hydroxyphenyl (**1**) and 2,4-dihydroxyphenyl (**3**, resorcinol moiety) substituents also exerted potent tyrosinase inhibition. These results indicate that the 4-substitueted resorcinol moiety importantly confers tyrosinase inhibiting activity.

The position and number of hydroxyl substituents on the phenyl ring significantly influenced the inhibitory activity against tyrosinase. However, the details of certain chemical structures attached to the one or two hydroxyphenyl moieties for tyrosinase activity remain unknown; this will be the focus of future research.

### 3.3. Enzyme Kinetic Studies of Compounds ***1**–**3***

To elucidate the type of enzymatic inhibition, a kinetic analysis was accomplished at several concentrations of the l-tyrosine substrate and inhibitors (**1**–**3**), according to the Lineweaver–Burk and Dixon plot methods (Table 3 and Figure 2). Kojic acid, a competitive tyrosinase inhibitor, was used as the standard [44]. Using the Lineweaver–Burk plot method, the lines representing inhibitors **1**–**3** intersected at the same point on the *y*-axis, demonstrating that *Km* increased with increasing concentrations of inhibitors **1**–**3**, while 1/*V_max_* did not change [33]. Compounds **1**–**3** showed competitive inhibition. These data suggest that compounds **1**–**3** are effective inhibitors that bind to the active site of the enzyme [7]. Furthermore, the Dixon plot is a graphical method [plot of 1/enzyme velocity (1/*V*) against inhibitor concentration (I)] used to determine the type of enzymatic inhibition and the dissociation or inhibition constant (*K*_ic_) for the enzyme-inhibitor complex [33].

The *K*_ic_ (≈*K*_i_) values for compounds **1**–**3** were 16.55, 3.21, and 3.01 µM, respectively, against mushroom tyrosinase (Table 3 and Figure 2). As the *K*_ic_ value represents the concentration required to combine the inhibitor with an enzyme, compounds with lower *K*_ic_ values were generally more effective tyrosinase inhibitors; this is an important requisite for the development of preventive and therapeutic agents.

### 3.4. Docking Simulation and of Ligands against Tyrosinase

Based on the in vitro results, compound **1**–**3**—docking complexes were evaluated to understand their binding conformation within the active site of the tyrosinase. To predict the binding site of the potent compounds **1**–**3**, molecular docking simulations were performed using the open-source programs for AutoDock4.2 [36] and AutoDock Vina [47] along with AutoDockTools, a graphical user interface compliment to the AutoDock software suite. AutoDock 4.0 uses a semi-empirical free energy force field to predict the binding free energies of protein–ligand complexes of a known structure and the binding energy for both bound and unbound states [36]. AutoDock vina was developed as a successor to AutoDock 4.2 and provides more rapid and accurate results with less direct investment required by the end user [48].

The most important factor involved in tyrosinase inhibition are coordinated with Cu ions, which played an important role in the activity. The binding energy, number of interaction residues of the compounds, and the reference compound kojic acid (competitive inhibitor) (Figure 3B) [49,50] are summarized in Table 4 and Figure 3A. The most potent compound **2** was stabilized by HIS85-, HIS244-, ASN260-, HIS263-, PHE264-, MET280-, GLY281-, SER282-, VAL283-, and GLU322-interacting (hydrogen/hydrophobic) residues (Figure 3D,G). Similarly, compound **1-** or compound **3**-tyrosinase complexes interacted with HIS61, HIS85, HIS259, HIS263, PHE264, ARG268, GLY281, SER282, VAL283, ALA286, and PHE292 residues, as shown in Figure 3C,F for compound **1**; and Figure 3E,H for compound **3**. The residues of active site of tyrosinase are HIS61, HIS85, HIS94, His259, HIS263, VAL283, and HIS296, detailed results for searching active site of tyrosinase have been previously reported [51], which also partially observed in our compounds **1**–**3**. The binding energies of –6.73/–6.3, –6.92/–6.5, –6.35/–6.4 kcal/mol were consumed by compounds **1**–**3**, as determined using the AutoDock 4.2 and AutoDock Vina programs. The binding scores of kojic acid were –4.21/–5.6 kcal/mol, indicating that compounds **1**–**3** may be bind to tyrosinase with stronger affinity than kojic acid. The basic skeleton of the synthesized compounds was similar; therefore, no significant difference in energy value was observed between the docking results. Data from the literature have also confirmed the importance of HIS259 and HIS263 residues in bonding with other tyrosinase inhibitors, supporting our docking results [51,52,53].

### 3.5. Cell Viability of Compounds ***1**–**3***

The anti-melanogenic activity of compounds **1**–**3**, which were the most effective in B16F10 cells, was tested in a cell-based in vitro model. The EZ-Cytox assay was used to estimate cell viability. Cells were treated with various concentrations (5–200 µM) of compounds **1**–**3** and trolox to examine whether these compounds exhibited cytotoxic effects on melanocytes. The results indicated that compounds **1**–**3** and trolox did not significantly affect cell viability at concentrations up to 20 µM compared with the vehicle control cells for 48 h (Figure 4). Consequently, 20 µM sample concentrations of compounds **1**–**3** and trolox were used for further experiments.

### 3.6. ROS Scavenging Activity

ROS may regulate melanogenesis in melanoma cells [20,54,55]. The principle of the intracellular ROS assay is that DCF-DA spreads through the cell membrane and is enzymatically hydrolyzed to DCF by esterase; DCFH then responses with ROS to yield DCF [56]. The ROS scavenging activity of compounds **1**–**3** was investigated in vitro by using fluorescence probes, (DCF-DA) for ROS. SIN-1 is used as an NO donor [57,58]. As demonstrated in Figure 5A, compound **2** exerted potent scavenging activity against ROS, with an IC_50_ value of 2.36 ± 0.48 µM, comparable with that of the reference control, trolox, which had an IC_50_ value of 13.25 ± 0.73 µM. However, compounds **1** and **3** did not present any activity at the tested concentrations at ≥50 µM. Furthermore, as shown in Figure 5B, treatment with SIN-1 significantly increased ROS generation compared to that in untreated B16F10 cells. Pretreatment with compound **2** reduced SIN-1-induced ROS generation in a dose-dependent manner. These results indicated that compound **2** attenuated the SIN-1-induced increase in ROS activity and intracellular ROS levels. Structure comparison of the benzimidaxothiazolone skeleton and ROS scavenging activities reveals that a *ortho*-dihydroxy groups of the benzylidene plays a crucial role in the observed scavenging activity. The importance of catechol moiety (3,4-dihydroxy group) for ROS scavenging activity was reported in previous studies in the literature [59,60].

### 3.7. Melanin Contents and Cellular Tyrosinase Assay

α-MSH and IBMX increase the levels of intracellular cAMP, which induces melanogenesis [9,61,62]. Melanin contents were measured after pretreating the B16F10 cells with different concentrations (5, 10, and 20 µM) of the compounds **1**–**3** for 3 h, followed by 48 h of α-MSH and IBMX treatment. The melanin contents increased to 134.78 ± 2.95%, 137.96 ± 3.02%, and 134.78 ± 2.95% (Figure 6A–C, respectively) following treatment with α-MSH and IBMX. As shown in Figure 6B, compound **2** led to a dose-dependent decrease in the intracellular tyrosinase activity to 116.75 ± 3.52%, 100.52 ± 2.08%, and 78.80 ± 3.66%. Kojic acid at 20 µM reduced the content to 128.53 ± 2.95%. In parallel to the enzyme assay, compound **1** reduced the melanin content at 20 µM (Figure 6A), and compound **3** increased the melanin contents in a dose-dependent manner (Figure 6C). To determine the inhibitory potency of compounds **1**–**3** in the cellular model system, the inhibitory effect on the tyrosinase activity of B16F10 cells treated with 5 µM α-MSH and 200 µM IBMX were examined. The level of intracellular tyrosinase after α-MSH and IBMX treatment was 347.03 ± 7.97%, 356.28 ± 27.70%, and 341.30 ± 26.76% (Figure 6D–F, respectively). However, cellular tyrosinase activity decreased in a concentration-dependent manner upon exposure to compounds **1**–**3**. In both melanin contents and cellular tyrosinase inhibitory assay, compound **2** exhibited potent inhibitory activities, indicating that the presence of 3,4-dihydroxy (catechol moiety) substituent of benzimidaxothiazolone derivatives influence inhibitory activity. These results suggest that compound **2** is a promising candidate therapeutic anti-melanogenesis agent among benzimidazothiazolone derivatives.

### 3.8. Effects of Compound ***2*** on Melanogenesis-Related Signaling

It is important to determine whether the inhibition of melanin synthesis by compound **2** is related to melanogenesis signaling and gene expression. To investigate whether compound **2** could affect the melanogenic signaling of tyrosinase, the expression of tyrosinase, phosphorylated CREB and PKA was determined with Western blot analysis using cytosolic factions from the cell lysates of α-MSH- and IBMX-induced B16F10 cells. As shown in Figure 7A,B, treatment with α-MSH and IBMX significantly upregulated these proteins; however, compound **2** decreased their expression on these proteins in a dose-dependent manner. MITF, a leucine zipper transcription factor, activates the expression of multiple genes encoding enzymes involved in the conversion of tyrosine into melanin, resulting in increased levels of these melanin-producing proteins [63,64]. Expression of MITF in α-MSH- and IBMX-induced B16F10 cells was upregulated, while compound **2** significantly suppressed MITF levels in the nuclear faction from total cell lysate (Figure 7C). The skin is site of oxidative stress due to exposure to daylight and environmental oxidizing pollutants. It is known that ROS stimulates α-MSH- and IBMX-induced production in melanoma cells leading to melanogenesis by upregulating MITF, a transcription factor that induces tyrosinase gene expression [65,66]. Since oxidative stress has been shown to activate MITF through MAPK, we investigated whether compound **2** controls MAPK signaling [67,68]. The results shown in Figure 7D,E revealed that compound **2** decreased the levels of phosphorylated JNK and ERK in a concentration-dependent manner. However, the levels of phosphorylated p38 were not affected by treatment with compound **2**. In our study, both p-JNK and p-ERK were decreased by compound **2**. Thus, our results suggest that compound **2**, which suppresses the phosphorylated PKA and MAPK signaling pathways and MITF-mediated melanogenic enzyme expression, has potential as an anti-melanogenic agent.

The effects of compound **2** on the expression of mRNA melanogenic factors were evaluated in α-MSH- and IBMX-induced B16F10 cells. Total mRNAs were extracted from cells treated with α-MSH and IBMX in the absence or presence of compound **2**. Tyrosinase, TRP1 and TRP2 are key enzymes involved in melanin biosynthesis [64]. During melanin synthesis, tyrosine undergoes tyrosinase-dependent conversion, which is catalyzed by dapaquinone and subsequently by dopachrome. TRP2 converts dopachrome to 5,6-dihydroxyindole-2-carboxylic acid (DHICA), whereas TRP1 oxidizes DHICA to a carboxylated indole-quinone, which is eventually converted into melanin [12,68,69]. The mRNA levels of tyrosinase, TRP1, TRP2, and MITF were quantified with real-time qPCR (Figure 7F). Compound **2** significantly inhibited the expression of the MITF and downstream enzymes, such as tyrosinase, TRP1, and TRP2. These results indicate that compound **2** might be an effective hypopigmenting agent with implications in various dermatologic hyperpigmentation disorders, such as freckles and melasma, and has useful effects in whitening cosmetic agents.

## 4. Conclusions

To determine whether benzimidazothiazolone with a (*Z*)-β-phenyl-α,β-unsaturated carbonyl scaffold plays an important role in tyrosinase inhibition, 13 derivatives were synthesized. The mushroom tyrosinase inhibitory assay revealed that three benzimidazothiazolone derivatives (compounds **1**–**3**) with 4-, 3,4-, and 2,4-dihydroxyls on the phenyl ring of the benzimidazothiazolone scaffold, respectively, inhibited tyrosinase significantly more than the positive control (kojic acid). Enzymatic kinetic studies indicated that compounds **1**–**3** were competitive inhibitors. Docking simulation of compounds **1**–**3** predicted that they bind more strongly to the active site of tyrosinase than kojic acid. In vitro experiments showed that compounds **1**–**3** inhibited cellular tyrosinase inhibitory activity in a dose-dependent manner. Compound **2** scavenged ROS and inhibited melanin biosynthesis downregulating the PKA/CREB and MAPK signaling pathways, leading to suppression of MITF expression and tyrosinase (Figure 8). This is the first report of benzimidazothiazolone with a (*Z*)-β-phenyl-α,β-unsaturated carbonyl scaffold for anti-tyrosinase and anti-melanogenesis effects to the best our knowledge.

In the present study, it was found that compound **2** suppressed intracellular ROS level in a dose-dependent manner. Compound **2** appears to be a potential whitening agent that might protect melanoma cells from oxidative injury. Hence, compound **2** has potential for use as a novel dermatological anti-melanogenesis agent and an effective skin-whitening agent. Although many synthetic inhibitors exhibited remarkable tyrosinase inhibitory activity, only a few of them showed melanogenesis inhibition activity in cell-based or skin models. Therefore, further studies may be required to assess the potential for developing safe products of skin-whitening agents by in vivo studies [70], such as zebra fish, and may have practical applications for humans.

## Data Availability

Data is contained within the article and Appendix A.

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
