# Peer review of "New Benzimidazothiazolone Derivatives as Tyrosinase Inhibitors with Potential Anti-Melanogenesis and Reactive Oxygen Species Scavenging Activities"

_antioxidants, 2021, doi:10.3390/antiox10071078_

Round 1
Reviewer 1 Report
Effect of compounds 1 – 3 on the viability of B16F10 cells should be modified, in particular increasing the concentrations range and using as the same time a positive ctrl, able to reduce the cell viability. Trolox should be used also for cell viability assay.
Conclusion section should be modified in order to put more emphasis on the real application of the compounds.
Author Response
Reviewer 1
Comments and Suggestions for Authors
- Effect of compounds 1 – 3 on the viability of B16F10 cells should be modified, in particular increasing the concentrations range and using as the same time a positive ctrl, able to reduce the cell viability. Trolox should be used also for cell viability assay.
Answer: Thank you for your suggestion. Following reviewer's comments, we determined the cell viability of compounds 1-3 and trolox at maximum concentrations of 200 µM with the vehicle control for 48 h in B16F10 cells. And we revised the Figure 4 in Results and Discussion section in the manuscript.
- Conclusion section should be modified in order to put more emphasis on the real application of the compounds.
Answer: Thank you for your kind suggestion. We revised and included the application of the benzimidazothiazolone derivatives in the Conclusion section and the sentence noted with red color.
Reviewer 2 Report
In this work, the authors present a research conducted on thirteen (Z)-2-(substituted benzylidene) benzimidazothiazolone analogues which were synthesized and evaluated for their inhibitory activity against mushroom tyrosinase. From these, the mushroom tyrosinase inhibitory assay revealed that three benzimidazothiazolone derivatives (compounds 1 – 3) with 4-, 3,4-, and 2,4-dihydroxyls on the phenyl ring of the benzimidazothiazolone scaffold, respectively, inhibited tyrosinase significantly more than the positive control (kojic acid). According to their in vitro experiments, compound 2 scavenged ROS and inhibited melanin biosynthesis downregulating the PKA/CREB and MAPK signaling pathways, leading to suppression of MITF expression and tyrosinase activity and, compound 2 has potential for use as a novel dermatological anti-melanogenesis agent and an effective skin whitening agent.
The manuscript is good in quality with clear description of the research and results.
Recommendation: Accepted in the present form.
Suggestion: In the Abstract section the first phrase “As part of our continued work on potent tyrosinase inhibitors, this study aimed to evalu- 10 ate the anti-melanogenic effects of benzimidazothiazolone derivatives.” to be moved at the end of the paragraph Introduction.
Author Response
Reviewer 2
Comments and Suggestions for Authors
In this work, the authors present a research conducted on thirteen (Z)-2-(substituted benzylidene) benzimidazothiazolone analogues which were synthesized and evaluated for their inhibitory activity against mushroom tyrosinase. From these, the mushroom tyrosinase inhibitory assay revealed that three benzimidazothiazolone derivatives (compounds 1 – 3) with 4-, 3,4-, and 2,4-dihydroxyls on the phenyl ring of the benzimidazothiazolone scaffold, respectively, inhibited tyrosinase significantly more than the positive control (kojic acid). According to their in vitro experiments, compound 2 scavenged ROS and inhibited melanin biosynthesis downregulating the PKA/CREB and MAPK signaling pathways, leading to suppression of MITF expression and tyrosinase activity and, compound 2 has potential for use as a novel dermatological anti-melanogenesis agent and an effective skin whitening agent.
The manuscript is good in quality with clear description of the research and results.
Recommendation: Accepted in the present form.
Suggestion: In the Abstract section the first phrase “As part of our continued work on potent tyrosinase inhibitors, this study aimed to evalu- 10 ate the anti-melanogenic effects of benzimidazothiazolone derivatives.” to be moved at the end of the paragraph Introduction.
Answer: Thank you for your kind evaluations. Following reviewer's comments, the first paragraph of Abstract section has been moved to the end of the paragraph in Introduction section.

Reviewer 3 Report
Manuscript antioxidants-1265677 is a continuation of work on tyrosinase inhibitors made by the research group during the last years and in present project 13 benzimidazothiazolone derivatives were prepared and tested. The work is correctly designed, manuscript is clearly written, methodology is presented with all needed details.
1. I was wondering why the authors decided to submit the manuscript to the “Antioxidants” journal as the problems directly relevant to antioxidant activity, ROS scavenging and even indirect antioxidant action of the studied compounds are on the margin of the main subject. Then, I looked for a connection between inhibitory activity and ROS scavenging of selected derivatives (Structure-Activity Relationship), and indeed, I am convinced that the subject is within the scope of the “Antioxidants”. The part of the discussion section on page 17, lines 527-536 informs that “oxidative stress stimulates melanogenesis by upregulating MITF….” (with reference 51, ie., a paper published by the same group 3 years ago) but the Authors do not provide a deeper explanation, as I would expect more detailed description how ROS and ROS scavenging are connected with up-regulation/down-regulation of MITF. At this version of ms. the ROS scavenging by compound 2 is a well documented correlation but more careful discussion is recommended to clear the problem (for example, comparison with literature data of anti-ROS activity of other tyrosinase inhibitors with and without catechol moiety). The observation that compound with catechol moiety is a ROS-scavenging agent, is rather not surprising, but how to explain that compounds 3 and 4 (non catechols) are also potent inhibitors of tyrosinase?
Besides this general comment I suggest to compress the manuscript:
2. The experimental details of synthesis and identification of all prepared compounds can be moved to Supporting Information as not essential for the main text of a paper (short description on the beginning of Results &Discussion section is enough)
3. Table 5 is not necessary: it contains IC50 parameters for comp 2 and for Trolox (already included in the manuscript text) whereas IC50 for two other compounds are >100 uM, that can be also described in the text.
4. Sixteen among 72 references is from the same group (22%). Are all those references essential for the current manuscript?
5. Abbreviation SIN-1 appears in line 468, but first time this abbreviation is used in line 264.
6. Figure 8: All processes are marked as down-regulated by compound 2, why the “up” arrow is also included with a description “Up-regulated by compound 2”?
Author Response
Reviewer 3
Comments and Suggestions for Authors
Manuscript antioxidants-1265677 is a continuation of work on tyrosinase inhibitors made by the research group during the last years and in present project 13 benzimidazothiazolone derivatives were prepared and tested. The work is correctly designed, manuscript is clearly written, methodology is presented with all needed details.
- I was wondering why the authors decided to submit the manuscript to the “Antioxidants” journal as the problems directly relevant to antioxidant activity, ROS scavenging and even indirect antioxidant action of the studied compounds are on the margin of the main subject. Then, I looked for a connection between inhibitory activity and ROS scavenging of selected derivatives (Structure-Activity Relationship), and indeed, I am convinced that the subject is within the scope of the “Antioxidants”. The part of the discussion section on page 17, lines 527-536 informs that “oxidative stress stimulates melanogenesis by upregulating MITF….” (with reference 51, ie., a paper published by the same group 3 years ago) but the Authors do not provide a deeper explanation, as I would expect more detailed description how ROS and ROS scavenging are connected with up-regulation/down-regulation of MITF.
At this version of ms. the ROS scavenging by compound 2 is a well documented correlation but more careful discussion is recommended to clear the problem (for example, comparison with literature data of anti-ROS activity of other tyrosinase inhibitors with and without catechol moiety). The observation that compound with catechol moiety is a ROS-scavenging agent, is rather not surprising, but how to explain that compounds 3 and 4 (non catechols) are also potent inhibitors of tyrosinase?
Answer: Yes, although the ROS scavenging activity of benzimidazothiazolone derivatives 1-3 were performed in cell-free in vitro (Fig. 5A) and in cell-based experiments (Fig. 5B), respectively, we desired to report for the first time in the journal Antioxidant the potent anti-melanogenesis and ROS scavenging activity of benzimidazothiazolone derivatives with substituent-benzylidene.
Sorry to confuse you. Although compound 2 is a potent mushroom tyrosinase inhibitor, this report did not fully explain the correlation between SIN-1-induced intracellular ROS scavenging and tyrosinase inhibitory activity. However, Compound 2 exerts to be a potential whitening agent that might protect melanoma cells from oxidative damage. Therefore, we would like to propose that compound 2 is presented as a ROS scavenger and an melanogenesis inhibitor.
We also added several papers on ROS-mediated up-regulation by MITF and carefully assembled them in the Results and Discussion section. In addition, we added to the literature on the correlation of compound 2 with ROS scavenging and tyrosinase inhibitors in the 3.6. Results and Discussion sub-section.
Compound 3 with 2,4-dihydroxy group (resorcinol moiety) also has a potent tyrosinase inhibitory activity. It has been reported in several literatures and also verified in this study. Compound 4 bearing 3-methoxy-4-hydroxy groups was moderate in its inhibitory activity against tyrosinase. These two compounds 3 and 4 also exert the potential tyrosinase inhibition and was performed for docking simulation with three programs (Autodock Vina, Autodock 4, Dock6) (data not shown). With the above contents, although both compound 3 and compound 4 were not having a catechol moiety in benzylidene, but we expected that the position of functional group such as hydroxyl, methoxy, and ethoxy are important for tyrosinase inhibitory activity. In addition, although docking simulation results predicted binding energy and information of adjacent amino acids, it is supposed that the prediction results of molecular simulation using several docking programs can also serve as sufficient supporting data to predict a strong tyrosinase inhibitors.
Besides this general comment I suggest to compress the manuscript:
Answer: Thank you for your suggestion. For the summarized the manuscript, we transfer the procedure of general synthesis for the compound as well as general experimental methods. And references have been reduced.
- The experimental details of synthesis and identification of all prepared compounds can be moved to Supporting Information as not essential for the main text of a paper (short description on the beginning of Results &Discussion section is enough)
Answer: We moved the synthesis and identification of all compounds part to create a Supporting information file.
- Table 5 is not necessary: it contains IC50 parameters for comp 2 and for Trolox (already included in the manuscript text) whereas IC50 for two other compounds are >100 uM, that can be also described in the text.
Answer: Table 5 has been deleted.
- Sixteen among 72 references is from the same group (22%). Are all those references essential for the current manuscript?
Answer: We will keep only essential references and deleted unnecessary references.
- Abbreviation SIN-1 appears in line 468, but first time this abbreviation is used in line 264.
Answer: We corrected the position of the abbreviation of SIN-1.
- Figure 8: All processes are marked as down-regulated by compound 2, why the “up” arrow is also included with a description “Up-regulated by compound 2”?
Answer: Following reviewer’s comments, we carefully modified the Figure 8 in the manuscript.
Round 2
Reviewer 1 Report
I suggest to accept the ms